# Comparative Transcriptomics Revealed *Physalis floridana* Rydb. Influences on the Immune System of the 28-Spotted Ladybird Beetle (*Henosepilachna vigintioctopunctata*)

**DOI:** 10.3390/plants13192711

**Published:** 2024-09-27

**Authors:** Xianzhong Wang, Liwen Guan, Tianwen Wang, Liuhe Yu, Shuangle Wang, Biner He, Bin Tang, Jiangjie Lu

**Affiliations:** 1College of Life and Environmental Sciences, Hangzhou Normal University, Hangzhou 311121, China; wangxz9264@163.com (X.W.); guanliwen1010@163.com (L.G.); 2021210315289@stu.hznu.edu.cn (T.W.); 2021210315167@stu.hznu.edu.cn (L.Y.); 2021210315186@stu.hznu.edu.cn (S.W.); 2021210315068@stu.hznu.edu.cn (B.H.); 2Zhejiang Provincial Key Laboratory for Genetic Improvement and Quality Control of Medicinal Plants, Hangzhou Normal University, Hangzhou 311121, China

**Keywords:** *Henosepilachna vigintioctopunctata*, *Physalis floridana* Rydb., transcriptomics, immune system, Toll and IMD signal pathway

## Abstract

*Physalis floridana* Rydb., a member of the Solanaceae family, is renowned for its diverse secondary metabolites, including physalins and withanolides. The 28-spotted ladybird beetle (*Henosepilachna vigintioctopunctata*) is a notorious pest severely damaging Solanaceous crops. This study demonstrates that *P. floridana* Rydb. significantly impacts on the development and reproductive suppression of *H. vigintioctopunctata*. A comparative transcriptome analysis was performed by feeding *H. vigintioctopunctata* larvae on *P. floridana* Rydb., *Solanum nigrum* L., *Solanum tuberosum* L., and *Solanum lycopersicum* L. The results reveal that larvae fed on *P. floridana* Rydb. exhibit numerous differentially expressed genes, which are notably enriched in pathways related to energy metabolism, immunity, and detoxification. These functions and pathways are less enriched in larvae fed by other hosts. Weighted Gene Co-expression Network Analysis (WGCNA) indicates that feeding on *P. floridana* Rydb. influences the expression of specific genes involved in the Toll and IMD signaling pathways, impacting the immune system of *H. vigintioctopunctata* larvae. This study provides transcriptomic insights into larval responses to different diets and suggests that the effect of *P. floridana* Rydb. on the immune system of *H. vigintioctopunctata* is a key defense mechanism against herbivores.

## 1. Introduction

The coevolution between plants and insects is a classic example in evolutionary biology, with plants developing a diverse array of secondary metabolites over 400 million years to defend against herbivory [1]. Utilizing extracts or secondary metabolites from plants with pesticidal potential to control pests is a long-standing research direction and an important strategy for environmentally friendly pest management. Since the U.S. Environmental Protection Agency approved neem-based pesticides as low-risk botanical pesticides in 1990, the field of plant-derived pesticides has seen renewed interest. Besides the United States, regions such as Europe, China, Brazil, and India have approved and widely marketed plant-based pesticides [2,3]. With increasing environmental standards and demand for green food, the market for plant-derived pesticides is rapidly growing and promising. Discovering new insect-resistant plants and active ingredients remains a crucial research focus for the development of plant-derived pesticides [4]. Moreover, companion planting, including trap cropping, leverages vegetative diversification to manage insect pests by attracting them away from main crops and conserving natural enemies, though optimal systems remain a topic of ongoing research. Despite the lack of consensus on the best practices, trap crops offer a promising approach by serving as both a diversion for pests and a refuge for beneficial organisms, potentially enhancing integrated pest management strategies [5,6].

Plants in the genus *Physalis* are important members of the Solanaceae family, known for their inflated, lantern-like calyces [7,8]. Beyond their distinctive calyxes, the rich repository of steroidal secondary metabolites is also a key reason for the extensive attention given to *Physalis* species [9]. Current research and application of *Physalis* plants focus primarily on their roles in traditional medicine and the bioactivities of important secondary metabolites, such as physalins and withanolides, which include antitumor, anti-inflammatory, antimalarial, and antimicrobial properties [10,11,12,13,14]. Research on the isolation of physalins, agricultural cultivation of *Physalis* species, and the biological regulation of secondary metabolite synthesis has also been conducted. A few studies have reported the insecticidal activity of *Physalis* species, including the feeding deterrent effects of physalins, as well as their impact on insect development and immune system damage [15,16,17]. *P. floridana* Rydb. is an important member of the *Physalis* genus, characterized by its stems and branches being covered with fine trichomes. Trichomes are a typical trait in plant defenses, with their bases serving as sites for the secretion of volatile organic compounds (VOCs) and the synthesis of secondary metabolites [18,19]. Like other members of the *Physalis* genus, *P. floridana* Rydb. is rich in steroidal secondary metabolites. Its more advanced genetic transformation system and artificial breeding methods make it a unique candidate for the development of the next generation of plant-based pesticides [20,21,22].

The 28-spotted ladybird beetle (*Henosepilachna vigintioctopunctata*) is a herbivorous insect in the family *Coccinellidae*, order Coleoptera, and is a notorious pest in horticulture. It is widely distributed in the subtropical regions of Southeast Asia and Australia, causing severe damage to crops such as eggplants, potatoes, tomatoes, and some cucurbit crops [23]. This pest predominantly feeds on plants from the Solanaceae family and occasionally infests those from the *Cucurbitaceae* family, rendering it a significant vegetable pest. Feeding on plants and their fruits leads to reduced crop quality and yield, and even plant death. However, the primary method of controlling *H. vigintioctopunctata* is through the use of chemical pesticides. Current exploration of green pest control focuses on the development of nucleic acid pesticides, primarily dsRNA-based products [24,25,26]. However, the long development cycle, susceptibility to degradation, and strict regulatory standards of nucleic acid pesticides make them unsuitable for green pest control [27,28]. How to achieve green pest control of *H. vigintioctopunctata* remains a critical challenge in vegetable pest management.

In previous studies, *Physalis* plants have been shown to possess the ability to affect the immune systems of insects, leading to their death and impacting reproduction. To explore whether it has similar effects on *H. vigintioctopunctata*, we fed *H. vigintioctopunctata* larvae with leaves from four plants, including *P. floridana* Rydb., and conducted transcriptome sequencing to investigate any differences and changes. In this study, we report the effectiveness of *P. floridana* Rydb. in controlling the important agricultural pest, the 28-spotted ladybird beetle, and explore the effects of different host plants on *H. vigintioctopunctata* using RNA-Seq. Our analysis reveals that the immune defense response of *H. vigintioctopunctata* fed on *P. floridana* Rydb. differs from that when fed on its natural host, *Solanum nigrum* L. Further enrichment analysis indicates that *P. floridana* Rydb. influences the Toll and IMD signaling pathways to induce humoral immunity in *H. vigintioctopunctata* in response to microbial threats. This study not only provides valuable insights into changes during the colonization process of polyphagous herbivorous insects on different hosts but also lays a theoretical foundation for the further development of plant-based pesticides using *Physalis* species.

## 2. Results

### 2.1. Feeding on P. floridana Rydb. Affects the Growth, Development, and Reproduction of H. vigintioctopunctata

All *H. vigintioctopunctata* larvae were divided into four groups and fed leaves from four different plants. The group fed *Solanum nigrum* L leaves is called the CK group; the group fed *P. floridana* Rydb. leaves is called the P106 group; the group fed *Solanum tuberosum* L. leaves is called the POT group; and the group fed *Solanum lycopersicum* L. leaves is called the TOM group. Among these four groups, the CK group serves as the control group, while the other three groups are experimental groups.

After feeding *H. vigintioctopunctata* on different plants, observations were made on third- and fourth-instar larvae, pupae, and adults. In terms of body size, larvae fed on P106 were smaller at the fourth instar stage compared to other groups, while the CK and Tom groups had the largest and most similar sizes. The Pot group had a slightly larger body size than the P106 group but smaller than the CK and Tom groups (Figure 1A). The number of eggs laid by female adults in the CK and P106 groups was statistically analyzed, and the number of eggs laid by adults fed on P106 was significantly lower than in the CK group. The average daily egg-laying rate per female in the CK group was 12.62, while in the P106 group, it was only 2.32 (Figure 1B). Dissection and observation of the ovaries of female adults in the CK, Pot, and P106 groups showed that on the third day after emergence the ovaries in all three groups were still developing, but the ovaries of the Pot and CK groups were more fully developed and better developed than those in the P106 group. On the fifth day after emergence, the ovaries of females in the CK and Pot groups were fully developed, with clear eggs visible, while the ovaries in the P106 group remained underdeveloped (Figure 1C).

### 2.2. Transcriptome Sequencing, Differential Gene Analysis, and qPCR Validation in H. vigintioctopunctata

A total of 78.71 Gb of data were sequenced using the DNBSEQ platform. After assembly and redundancy removal, 25,822 unigenes were obtained, with a total length of 28,032,351 bp, an average length of 1085 bp, an N50 of 1686 bp, and a GC content of 36.09%. Among the unigenes, those with sequence lengths of 200–300 bp were the most numerous (Figure 2A). As the sequence length increased, the number of unigenes decreased. The unigenes were then aligned and annotated against five functional databases. Ultimately, 18,214 (NR: 70.54%), 5668 (NT: 21.95%), 13,831 (SwissProt: 53.56%), 13,331 (KOG: 51.63%), and 13,922 (Pfam: 53.92%) unigenes received functional annotations (Figure 2B). TransDecoder identified 15,642 coding sequences (CDS). Additionally, 1043 simple sequence repeats (SSRs) were detected in 928 unigenes, and 1272 unigenes encoding transcription factors were predicted. Based on the unigene data, expression quantification was performed for twelve sample groups (Figure 2C), and principal component analysis (PCA) was conducted using the resulting expression profiles (Figure 2D). The PCA results show that the four sample groups are relatively concentrated within themselves, but separate between groups, indicating that there are significant differences between different sample groups, while samples within the same group are relatively similar.

After quantifying all genes, DESeq2 was used for normalization and differential expression analysis. Genes with a fold change greater than 1 and a *p*-value less than 0.05 were considered to have significant differential expression. Compared to the CK group, the P106 group had the highest number of differentially expressed genes (DEGs), totaling 3126, with 1783 DEGs significantly upregulated and 1343 DEGs significantly downregulated (Figure 3A). Compared to the CK group, the Pot group had only 1096 DEGs, with 639 DEGs significantly upregulated and 457 DEGs significantly downregulated (Figure 3B). In contrast, compared to the CK group, the Tom group had almost no differences, with only 31 DEGs, including 19 DEGs significantly upregulated and 12 DEGs significantly downregulated (Figure 3C). Among all the DEGs, 2522 were unique to the P106 group, 488 were unique to the Pot group, and 13 were unique to the Tom group. Among the shared DEGs, 592 were common to both the P106 and Pot groups, the highest number, while the Pot and P106 groups shared only 6 and 2 DEGs with the Tom group, respectively. Ten DEGs were found in all three comparison sets (Figure 3D).

After quantifying all genes, to verify the reliability of the transcriptome data, eight genes were randomly selected from all genes for RT-qPCR validation, using the CK group as the control. The relative expression levels were measured and compared with the fold change values. The results show that the RT-qPCR results are generally consistent with the fold change results, demonstrating a positive correlation (Figure 4).

### 2.3. GO Functional Enrichment and KEGG Pathway Enrichment Analysis Reveals Host Plant-Dependent Changes in H. vigintioctopunctata

After annotating and performing differential gene expression analysis on all sequenced genes, the differentially expressed genes (DEGs) obtained using the CK group as a control were used for GO functional enrichment analysis and KEGG pathway enrichment analysis to discover the patterns of change and differentially expressed functions in *H. vigintioctopunctata* under different host feeding conditions. Of all the enriched terms, the top 30 terms with the highest enrichment were visualized in Figure 5.

In the P106 vs. CK results, the GO functional enrichment analysis showed that the term with the highest enrichment was “antibacterial humoral response”, while the most significantly enriched term was “defense response to bacterium”. Other terms, such as “response to reactive oxygen species” and “isocitrate dehydrogenase (NAD+) activity”, suggest that the P106 group may have experienced cellular damage leading to an increase in reactive oxygen species compared to the CK group. The remaining terms were primarily focused on cell structure and energy metabolism-related functions (Figure 5A). In the KEGG pathway enrichment analysis, the term with the highest enrichment was “Collecting duct acid secretion”, and the most significantly enriched term was “Oxidative phosphorylation”. Notably, terms such as “Lysosome”, “Natural killer cell-mediated cytotoxicity”, and “Apoptosis” in the KEGG enrichment results indicate the possibility of cellular damage in P106-fed larvae. Additionally, some terms, such as “RNA polymerase”, “Protein digestion and absorption”, and “Ribosome”, are related to transcription and translation processes, while terms like “Drug metabolism—cytochrome P450”, “Drug metabolism—other enzymes”, and “Metabolism of xenobiotics by cytochrome P450” are related to detoxification (Figure 5B).

In the Pot vs. CK enrichment results, six terms reached 100% enrichment, including “cellular response to interferon-gamma”, “hydroxymethylglutaryl-CoA synthase activity”, “acetyl-CoA metabolic process”, “farnesyl diphosphate biosynthetic process, mevalonate pathway”, “negative regulation of plasminogen activation”, and “nuclear matrix”. However, the number of genes enriched in these terms was small. The most significantly enriched term was “structural constituent of cuticle”, with 127 genes enriched in this term (Figure 5C). The KEGG pathway enrichment results are shown in Figure 5D, with the highest and most significantly enriched term being “RNA polymerase”. Additionally, pathways such as “Oxidative phosphorylation” and “Thermogenesis”, which are related to energy metabolism, were also enriched.

In the Tom vs. CK enrichment results, the GO functional enrichment analysis showed weak significance, with the highest enrichment degree at 50%. The maximum −log10 (Q-value) was 1.33, and the significantly enriched terms included “phospholipase C-activating G protein-coupled receptor signaling pathway”, “electron transporter, transferring electrons within CoQH2-cytochrome c reductase complex activity”, “phosphoglycerate dehydrogenase activity”, and “mitochondrial ATP synthesis coupled proton transport” (Figure 5E). The KEGG pathway enrichment analysis results showed the highest enrichment degree of 4%, and none of the Q-values were significant, making them not meaningful for analysis (Figure 5F).

### 2.4. WGCNA Reveals the Link between P. floridana Rydb. Feeding and the Antimicrobial Response in H. vigintioctopunctata

Weighted gene co-expression network analysis (WGCNA) was performed on all differentially expressed genes. A gene co-expression network was established and clustered based on expression patterns, resulting in nine modules (Figure 6A). Modules were correlated with grouping characteristics, using the host plant fed on as a feature. The results showed that the module with the highest correlation to the CK group was MEgreen and MEred, with a correlation of 0.63, indicating a positive association; the module most associated with the Pot group was MEyellow, with a correlation of 0.75, indicating a negative association; the module most associated with the Tom group was MEblue, with a correlation of 0.59, also showing a positive association; notably, the module with the highest correlation to the P106 group was MEbrown, with a correlation of 0.88, showing a positive association and a *p*-value of 0.00014, which is extremely significant (Figure 6A).

To further investigate changes in the larvae of the P106 group, we conducted additional analysis on the brown module. In the scatterplot of gene significance (GS) versus Module Membership (MM) within the brown module, most genes appear in the upper left corner of the scatterplot and show significant correlation (correlation coefficient = 0.68, *p* = 5.4 × 10^−38^) (Figure 6C). A heatmap of all genes located in the brown module is shown in Figure 6D, where almost all genes were upregulated in the three replicates of the P106 group, while most were downregulated in the other three groups, further indicating that the gene expression changes in MEbrown are related to feeding on P106. GO analysis and KEGG analysis of the P106 group revealed enrichment results indicative of an association with immune responses.

Therefore, hub genes in MEbrown were selected based on MM values and GS values, yielding a total of 242 hub genes. Among these, 15 genes were found in the GO terms “defense response to bacterium” and “antibacterial humoral response”. Of these 15 genes, excluding those not annotated by KEGG, the remaining genes were annotated as part of the Toll pathway in the “Toll and IMD signaling pathway”. A network diagram was constructed using these 15 genes as hubs, including genes with a correlation weight greater than 0.25 (Figure 6E). The analysis of the network graph shows complex relationships among the hub genes, and all hub genes exhibited significantly increased expression levels in the P106 group compared to the CK group. Among the hub genes, *CL109.Contig1_All* did not have direct co-expression relationships with the other hub genes but had the most co-expression-related genes, some of which connected it to other hub genes.

### 2.5. Feeding on P. floridana Rydb. Affects the Toll Signaling Pathway of H. vigintioctopunctata, Leading to an Immune Response

The Toll pathway is crucial for arthropods, serving as an important route for recognizing pathogens and initiating immune responses. In insects, bacterial and fungal polysaccharide components are recognized, generating signals that are transmitted to the Toll receptor on the cell membrane. The Toll receptor then relays these signals into the cytoplasm, leading to the phosphorylation of the Cactus protein, which enters the ubiquitination pathway and is degraded. This degradation releases Dorsal or Dif, which can then enter the nucleus and activate the transcription of immune-related genes, thus triggering an immune response to combat infections.

In the WGCNA analysis conducted, we found that the hub genes obtained were enriched in the Toll pathway according to KEGG annotations. Compared to the CK group, the P106 group showed differential expression, with many members of the Toll pathway exhibiting significant changes in expression (Figure 7A in red). A heatmap was constructed using the FPKM values of all related genes annotated by KEGG across four groups. The results indicate that most of the differentially expressed genes have low expression levels, while only a few showed high expression levels, and there were differences between the groups. Among the four groups, the Pot group has the lowest gene expression levels for all highly expressed genes. In comparison to the CK group, the P106 group only shows slightly higher expression of Def, while the expression of the other genes was lower than in the CK group (Figure 7B).

## 3. Discussion

The transcriptome sequencing results indicated that feeding on *P. floridana* Rydb. led to substantial gene expression changes in *H. vigintioctopunctata* larvae. Through functional enrichment and pathway enrichment analyses, we found that the immune defense and detoxification responses triggered by feeding on *P. floridana* Rydb. were not enriched in other hosts. Energy metabolism, gene expression, and protein translation pathways were enriched in the P106 and Pot group but not in the Tom group. *Solanum nigrum* L., a primary wild host of *H. vigintioctopunctata*, contains solanine as its main secondary metabolite; however, long-term coexistence has rendered it almost ineffective against the beetles [29]. In contrast, potatoes and tomatoes, as important crop plants, have had their toxicity reduced or eliminated during human domestication and selection [30,31]. *P. floridana* Rydb. is a wild solanaceous plant that retains numerous secondary metabolites to resist various biotic stresses. Its rich secondary metabolites may be a key reason why *H. vigintioctopunctata* larvae need to activate more detoxification-related biological processes.

In weighted gene co-expression network analysis (WGCNA), we identified a significantly associated module, in which 15 highly expressed hub genes related to immunity were found and enriched in the Toll and IMD signaling pathways. Annotation revealed that these hub genes encode various antimicrobial peptides (Appendix A). In animals, the balance between immunity, growth, and reproduction is critical [32,33]. Stanley et al. propose that insect cellular immunity is not just a complement to humoral immunity but also plays a vital role in molting development and hypoxic stress [34]. Guo et al.’s research suggests that the Toll and IMD signaling pathways are not merely pathways for immune activation; bees can use them to regulate reactive oxygen species levels to select symbiotic bacteria [35]. These findings suggest that in insects, the immune system, particularly through the Toll and IMD signaling pathways, serves other important functions besides defending against external threats [36,37]. The transcriptome analysis in this study reveals changes in the *H. vigintioctopunctata* immune response, with a notable activation of humoral immunity. In this experiment, all plants were grown in the same environment to maximize consistency in their exposure to environmental factors, thereby reducing differences caused by pathogens on the plants. As a result, the activation of immune responses can be attributed to the plant hosts themselves rather than external pathogens. This explains why, in functional enrichment, energy metabolism, amino acid transport, and protein synthesis and degradation functions and pathways are highly enriched. The consumption of substances and energy in the immune system might be one of the key reasons for the meager egg-laying rate of adult beetles in the P106 group. However, biochemical level changes and molecular mechanisms require further validation and discovery.

*P. floridana* Rydb., as a member of the *Physalis* genus, is rich in physalins and withanolides. Barthel et al. confirmed that withanolides in *Physalis* plants have significant antibacterial properties and can stimulate the immune response in *Heliothis subflexa* [38], whereas in other insects, withanolides inhibit the immune response [39]. Physalins have been shown to modulate immunity and reduce inflammation at the cellular level and in mice [40,41,42]. They have been observed to have significant impacts on insect immune systems and symbiotic microbiota in *Rhodnius prolixus* [15,16]. In this study, *H. vigintioctopunctata* larvae in the P106 group exhibited significant changes in their immune system, possibly related to the presence of physalins and withanolides in *P. floridana* Rydb.

In conclusion, we present transcriptome data of *H. vigintioctopunctata* larvae reared under different host feeding conditions. Our data analysis revealed that *P. floridana* Rydb. influences the immune system of *H. vigintioctopunctata*, potentially affecting its growth, development, and reproduction.

## 4. Materials and Methods

### 4.1. Host Plant Cultivation

Seeds of *P. floridana* Rydb. (P106) were sourced from our laboratory, while *S. nigrum* seeds were collected in August 2022 from the campus of Hangzhou Normal University (120°02′ E, 30°30′ N). Potato (*S. tuberosum*) tubers came from Qingdao Jinmandi Potato Co., Ltd., Qingdao, China, and tomato (*S. lycopersicum*) seeds were purchased from Qingfeng Seed Industry Co., Ltd., Dezhou, China. All plants were grown in soil within a climate-controlled chamber maintained at 26 ± 1 °C and 70 ± 5% humidity, with a 16 h light–8 h dark cycle. All test plants were grown in the same room using identical soil to ensure that they were exposed to consistent environmental factors.

### 4.2. Inscts Rearing

*H. vigintioctopunctata* were collected from *S. nigrum* plant leaves on the campus of Hangzhou Normal University in August 2022 (120°02′ E, 30°30′ N) and reared in a controlled environment chamber at 28 ± 1 °C and 60 ± 5% humidity, under a 17 h light–7 h dark cycle. The larvae were housed in square plastic containers (10 cm × 5 cm × 5 cm) with small holes for ventilation. During the first and second instars, *S. nigrum* leaves served as food, while different host plant leaves were provided after the second molt. To ensure adequate nutrition, enough leaves were placed in the containers, which were replenished every 12 h. Adult beetles were raised in gauze cages (50 cm × 50 cm × 60 cm) and fed with two-month-old *S. nigrum* plants.

### 4.3. Statistical Analysis of Egg Production and Biological Image Capture

Pairs of female and male beetles, which were being fed *P. floridana* Rydb. (P106) leaves from the third instar, were placed in rearing boxes and continued to be fed P106 leaves. Fresh P106 leaves were provided every 24 h, and the number of eggs laid and the number of females present each day were recorded. If a female beetle died, no replacement was made; if a male beetle died, the box was replenished with the corresponding number of males to ensure there were always at least as many males as females. The control group was treated similarly. The adults were dissected and their ovaries were observed on the third and fifth days after eclosion. All images of the beetles and ovaries were captured using a Leica EZ4 HD Stereo Microscope (Leica Microsystems, Wetzlar, Germany).

### 4.4. Grouping Design and Sample Collection

The larvae were divided into four groups based on their host plants: those feeding on *P. floridana* Rydb. leaves were named the P106 group, those feeding on *S. nigrum* leaves were named the CK group, those feeding on *S. tuberosum* leaves were named the Pot group, and those feeding on *S. lycopersicum* leaves were named the Tom group.

Samples of larvae from each of the four groups were obtained 108 h after being fed on their respective host plants. Each group had three biological replicates, with each replicate consisting of three larvae to minimize individual variation. All samples were immediately frozen in liquid nitrogen after collection and stored at −80 °C. Once all samples were collected, they were shipped on dry ice to BGI Genomics Co., Ltd., Shenzhen, China, for RNA extraction, quality control, and sequencing.

### 4.5. Sequencing Platform, Data Library Construction and Analysis

The BGI Genomics DNBSEQ platform was used as the sequencing platform in this study, following the initial data filtering process carried out with SOAPnuke software (v1.4.0) to obtain clean reads [43]. The next step involved de novo assembly of the clean reads, with the removal of PCR duplicates to enhance assembly efficiency. The Trinity software (v2.0.6) was used for the assembly process, followed by clustering and redundancy removal of assembled transcripts using Tgicl to construct a comprehensive reference gene library. Subsequently, the clean reads were aligned to the reference gene sequences utilizing Bowtie2 (v2.2.5) to generate alignment results. To identify unigene candidate coding regions, the Transdecoder software (v3.0.1), as recommended by Trinity, was utilized. The process involved the initial extraction of the longest open reading frame, followed by predicting homologous protein sequences through Blast alignment against SwissProt and Pfam searches using Hmmscan to predict coding regions. The assembled unigenes underwent annotation in five major functional databases (NR, NT, SwissProt, Pfam, and KOG). All unigenes were aligned against the AnimalTFDB 2.0 database to predict transcription factors and provide annotations. Furthermore, clean reads were aligned to the reference gene library using Bowtie2, followed by the utilization of RSEM to calculate gene expression levels for each sample. Differential gene expression analysis between groups was then conducted using DESeq2 (1.22.2), with genes being considered differentially expressed when the *p*-value was less than or equal to 0.05 [44]. Based on GO and KEGG annotation results and official classifications, differentially expressed genes were categorized into functional and biological pathway categories. Enrichment analysis was performed using the phyper function in R software (R version 4.3.2) to calculate *p*-values, which were then corrected for false discovery rates (FDR). Functions with a Q-value ≤ 0.05 were considered significantly enriched. The top 30 terms with the smallest Q-values were selected for visualization using the ggplot2 package (v3.4.4) in R.

### 4.6. RT-qPCR Verification

RNA extraction was first performed on homogenized samples using the RNAiso Plus kit (Takara, Kyoto, Japan), following the manufacturer’s instructions. The quality and concentration of the extracted RNA were determined using a Nanodrop 2000 spectrophotometer (Thermo Fisher Scientific, Waltham, MA, USA), and the integrity was assessed by electrophoresis on a 1% agarose gel. The purified RNA was then stored at −80 °C for subsequent experiments. Following the RNA extraction, first-strand complementary DNA (cDNA) was synthesized using the PrimeScript RT reagent kit with gDNA Eraser (Takara, Kyoto, Japan) using the qualified RNA as a template, and the resulting cDNA was stored at −20 °C. The cDNA template was diluted threefold for quantitative PCR (qPCR). For qPCR analysis, *HvRPL13* was chosen as the internal reference gene [45], and specific qPCR primers were designed using Primer Premier 5 software and validated for specificity. The details of all primers are shown in Appendix A. The qRT-PCR was carried out in a 20 μL reaction volume using TB Green Premix Ex Taq II (Tli RNaseH Plus) (Takara, Kyoto, Japan) according to the manufacturer’s instructions. The qRT-PCR program included an initial denaturation step at 95 °C for 5 min, followed by 39 cycles of denaturation at 95 °C for 5 s and annealing/extension at 60 °C for 20 s. A melting curve analysis was performed by heating the reaction mixture to 65 °C for 5 s with continuous fluorescence monitoring to ensure amplification specificity. The relative gene expression levels were calculated using the 2^−ΔΔCT^ method. Finally, scatter plots and fitting curves were generated in Microsoft Office 2022 Excel software using the relative expression data and fold change values obtained from transcriptome analysis.

### 4.7. Weighted Gene Co-Expression Network Analysis

The WGCNA analysis was performed using the WGCNA shiny plugin in TBTool-II (version 2.110) [46]. In the first-time filter, the sample percentage was set to 0.9, and the expression cutoff to 1. For the second-time filter, the filter method was set to MAD, and the reserved genes’ Num. was set to 5000. The WGCNA power was manually chosen as 14, and all other parameters were left at their default settings. The resulting association network was visualized using Cytoscape (version 3.10.1).

### 4.8. Data Analysis

The spawning data were analyzed for significance using IBM SPSS Statistics 26 software and graphed using GraphPad Prism version 9.0 software.

## Figures and Tables

**Figure 1 plants-13-02711-f001:**
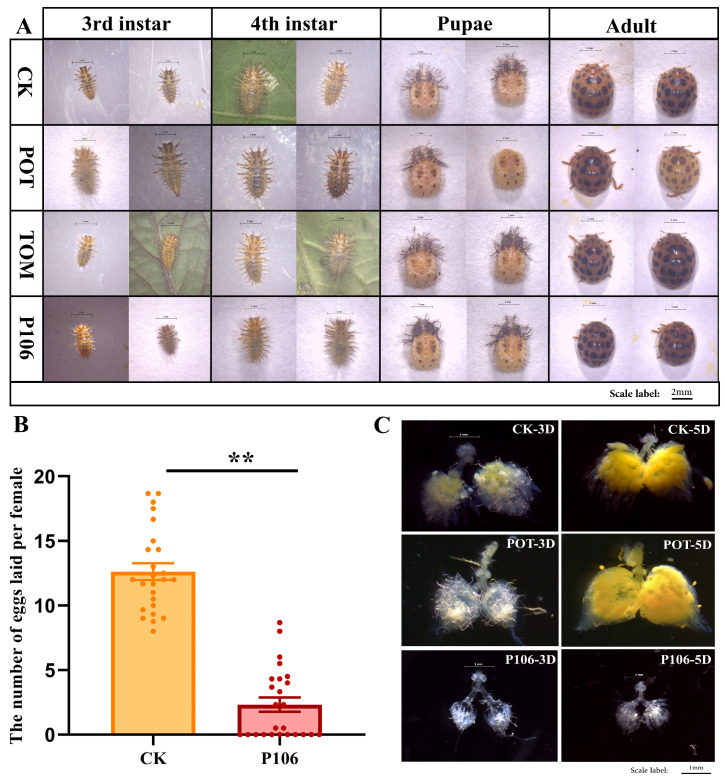
Feeding of *P. floridana* Rydb. influences the reproduction and development of *H. vigintioctopunctata*. (**A**) The images of larvae from four groups at the third instar, fourth instar, pupal stage, and adult stage were examined; (**B**) the average egg production of each female adult in the CK group and the P106 group, double asterisks (**) denote extremely significant differences, with each point representing an individual observation; (**C**) CK group, Pot group, and P106 female’s ovarian images after eclosion at 3 days and 5 days.

**Figure 2 plants-13-02711-f002:**
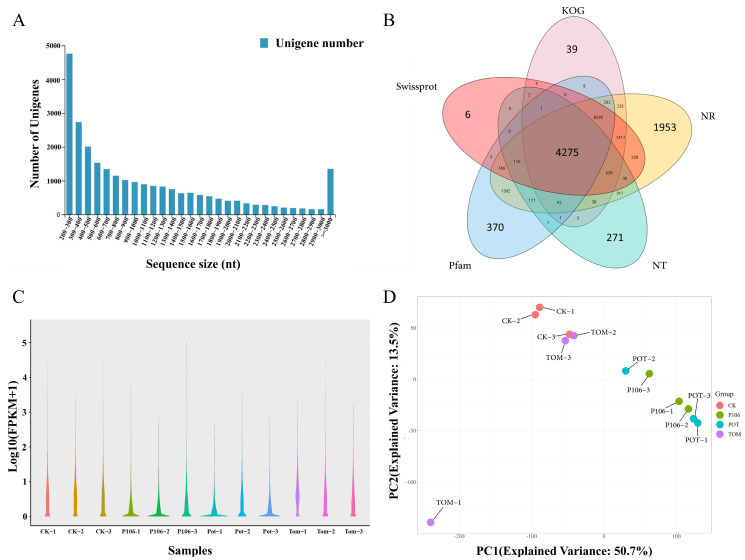
The assembly, annotation, quantification, and sample analysis results from the transcriptome sequencing. (**A**) The length distribution chart of all unigene sequences; (**B**) Venn plot showing the number of unigenes annotated in the five major databases; (**C**) Violin plot showing the distribution of expression levels across all samples; (**D**) principal component analysis (PCA) plot for the four groups of samples.

**Figure 3 plants-13-02711-f003:**
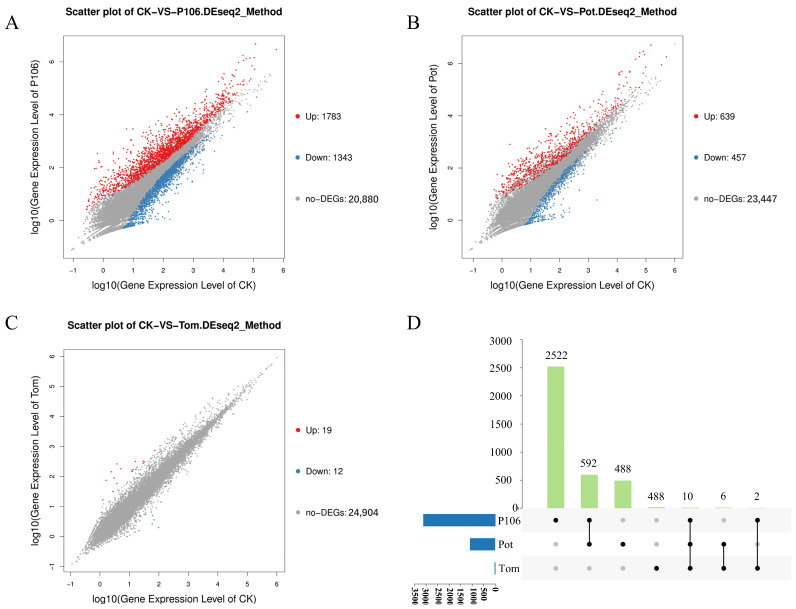
The number of differentially expressed genes (DEGs) in each group, with the CK group serving as the control. (**A**,**B**), and (**C**), respectively, show the distribution of DEGs in the P106, Pot, and Tom groups. Each point represents a gene, with red points indicating significantly upregulated genes, blue points representing significantly downregulated genes, and gray points denoting genes without significant changes. (**D**) is an upset plot for the three experimental groups, where the blue bars represent the number of unique genes in each group, and the green bars indicate the number of DEGs shared between the groups.

**Figure 4 plants-13-02711-f004:**
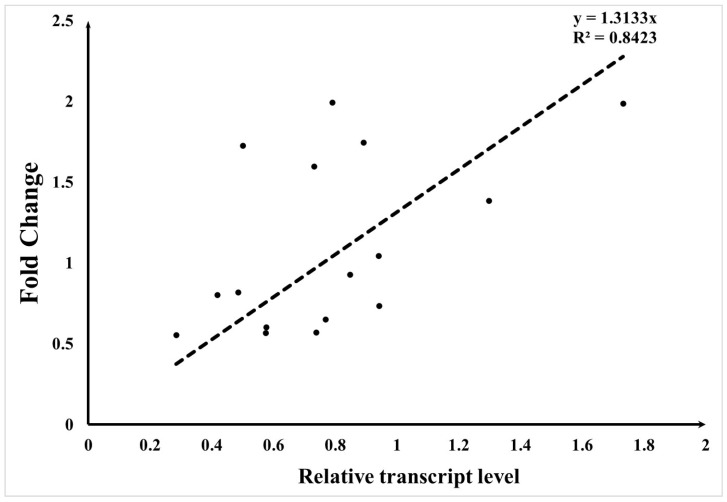
Transcriptome validation scatter plot. The *y*-axis represents the fold change value, and the *x*-axis represents the relative expression level. Each point represents a gene, and the dashed line indicates the fitted curve.

**Figure 5 plants-13-02711-f005:**
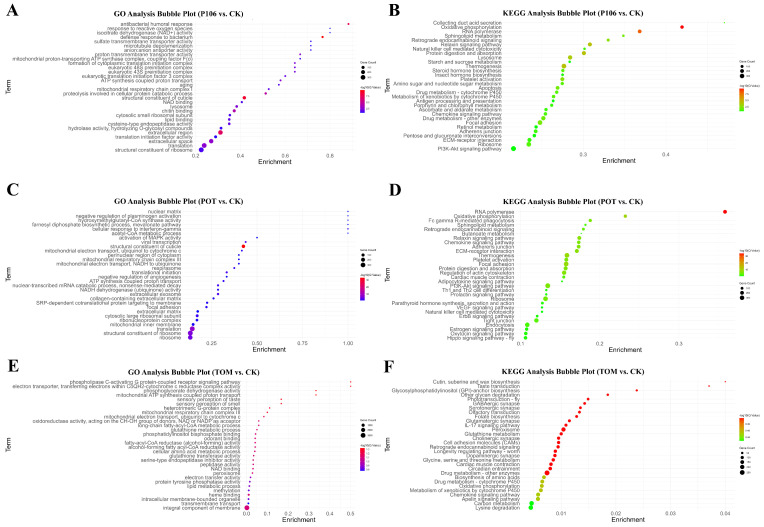
Bubble charts showing the GO functional and KEGG pathway enrichment results for each group. (**A**,**C**,**E**) represent the GO functional enrichment results for the P106, Pot, and Tom groups, respectively, while (**B**,**D**,**F**) represent the KEGG pathway enrichment results for the P106, Pot, and Tom groups, respectively.

**Figure 6 plants-13-02711-f006:**
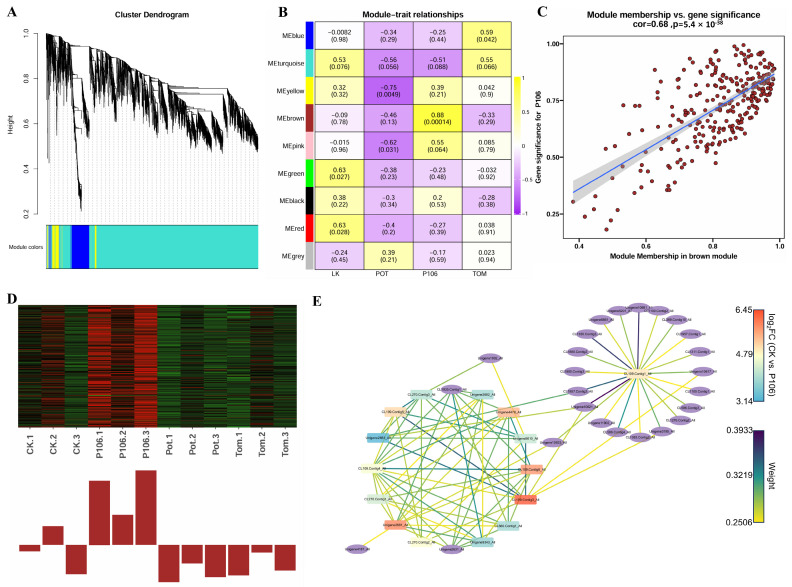
WGCNA analysis result. (**A**) The Cluster Dendrogram, representing the clustering results of the genes. Each color represents a module. (**B**) Heatmap showing the correlation between modules and different diets. Each cell contains a number representing the correlation coefficient, with positive numbers indicating positive correlation (yellow) and negative numbers indicating negative correlation (purple). The closer the absolute value is to 1, the higher the correlation. The numbers in parentheses represent the corresponding *p*-values, which indicate the statistical significance of the correlation. Each color on the vertical axis represents a module. (**C**) Scatter plot of MM vs. GS for the brown module. Each point represents a gene, with the *y*-axis indicating gene significance for the P106 group and the *x*-axis indicating Module Membership in the brown module (MEbrown). Genes closer to the upper right corner indicate a stronger association with the P106 group treatment. The blue line represents the functional curve fitted to these points, and the gray area represents the 95% confidence interval. (**D**) Heatmap of expression levels for all genes in the MEbrown module across 12 samples, with a bar chart below indicating their overall expression pattern. (**E**) Network graph of the 15 hub genes and their directly associated genes. Hub genes are represented by rounded rectangles, and associated genes by ellipses. The color of the hub genes indicates their expression levels, and the color of the edges represents the weight of the correlation between the nodes at either end.

**Figure 7 plants-13-02711-f007:**
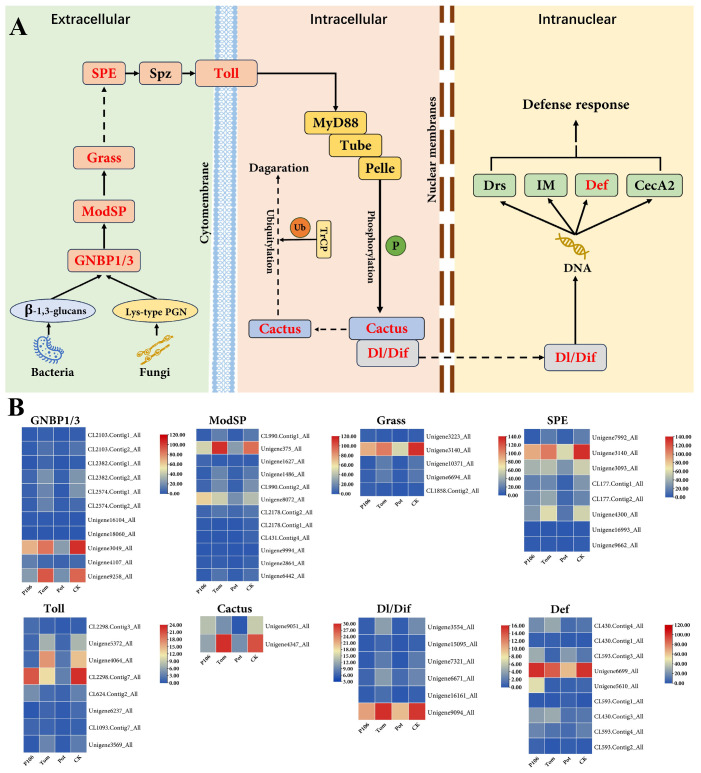
Changes in the Toll signaling pathway in *H. vigintioctopunctata* due to *P. floridana* Rydb. feeding. (**A**) Schematic diagram of the Toll signaling pathway. Each rectangle represents a protein, with proteins highlighted in red indicating that the corresponding gene’s expression level differs between the CK and P106 groups. (**B**) Heatmap of the expression levels of genes corresponding to differentially expressed proteins annotated in KEGG.

## Data Availability

The data that support the findings of this study are available on request from the corresponding author, [J.L.], upon reasonable request.

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
