# Peer review of "Comparative Transcriptomics Revealed Physalis floridana Rydb. Influences on the Immune System of the 28-Spotted Ladybird Beetle (Henosepilachna vigintioctopunctata)"

_plants, 2024, doi:10.3390/plants13192711_

Round 1

Reviewer 1 Report

Comments and Suggestions for Authors

This is a simple study that looks at the feeding response of an herbivore on particular species of plants. Overall the theoretical background is somewhat interesting as the authors make a case about pesticides and relate this to the feeding of insects on a plant. I think the connection here is very weak and could be improved. Maybe if the experiment was grinding up plants and making a spray but that appears to not be the case in this paper. I think this background could be revised to better reflect a more ecological approach. This would be a major re-write of the introduction but I think that would be appropriate as the background does not really match or frame the current study well.

The formatting calls for the methods at the end and it makes reading the results and discussion very hard as its not clear what was done by the time you read the results. I would re-write the results to make it more clear what was done and to simplify them overall for the reader.

The methods and the results make sense and are appropriate.

Line 59: More information on the breadth of foods, is it only restricted to the Solanaceae and Cucurbitaceae?

Line 72: The hypothesis and a short idea of the methods should be stated clearly here. This could vastly improve the readability of the manuscript.

Line 85: The lines of plants fed to insects is very confusing when presented in the journals format of having results before methods (instead of saying germ lines talk about the species of plants).  

Comments on the Quality of English Language

Many minor things that need to be addressed and some odd phrasing. 

Author Response

Reviewer 1:

Thank you for your thorough review. Your feedback is much appreciated. We have made detailed revisions to the manuscript based on your suggestions. In this revision, all changes are highlighted in blue font. Thank you once again for your comments on our work.

Comment 1: This is a simple study that looks at the feeding response of an herbivore on particular species of plants. Overall the theoretical background is somewhat interesting as the authors make a case about pesticides and relate this to the feeding of insects on a plant. I think the connection here is very weak and could be improved. Maybe if the experiment was grinding up plants and making a spray but that appears to not be the case in this paper. I think this background could be revised to better reflect a more ecological approach. This would be a major re-write of the introduction but I think that would be appropriate as the background does not really match or frame the current study well.

Reply: Thank you very much for your input. In this article, our initial intention was to apply secondary metabolites, such as physalins from Physalis floridana, to develop new types of pest control substance; therefore, our examples were about pesticides. Thus, we base our approach on the potential for developing secondary metabolites from P. floridana. Your suggestion is very meaningful, pointing out its potential role in ecological pest management. We have made some revisions in the 'Introduction' section to incorporate your suggestions. The specific changes are as follows (Line 41-47) :

Moreover, companion planting, including trap cropping, leverages vegetative diversi-fication to manage insect pests by attracting them away from main crops and con-serving natural enemies, though optimal systems remain a topic of ongoing research. Despite the lack of consensus on the best practices, trap crops offer a promising ap-proach by serving as both a diversion for pests and a refuge for beneficial organisms, potentially enhancing integrated pest management strategies.

Comment 2: The formatting calls for the methods at the end and it makes reading the results and discussion very hard as its not clear what was done by the time you read the results. I would re-write the results to make it more clear what was done and to simplify them overall for the reader.

Reply: Thank you for your suggestion. In this revision, we have directly clarified the grouping of the experiment in the Results section to make it easier to read. The specifics are as follows (Line98-103):

All H. vigintioctopunctata larvae were divided into four groups and fed leaves from four different plants. The group fed Solanum nigrum L leaves is called the CK group; the group fed P. floridana leaves is called the P106 group; the group fed Solanum tuberosum L. leaves is called the POT group; and the group fed Solanum lycopersicum L. leaves is called the TOM group. Among these four groups, the CK group serves as the control group, while the other three groups are experimental groups.

Comment 3: The methods and the results make sense and are appropriate.

Reply: Thank you for your recognition of our work.

Comment 4: Line 59: More information on the breadth of foods, is it only restricted to the Solanaceae and Cucurbitaceae?

Reply: Yes, your understanding is correct. H. vigintioctopunctata is a relatively host-specific pest that primarily damages plants in the nightshade family (Solanaceae). In some cases, it may also feed on plants in the cucumber family (Cucurbitaceae). In this revision, we have added a sentence describing its feeding habit to make this pest more accessible to the reader. The details are as follows (Line70-72):

This pest predominantly feeds on plants from the Solanaceae family and occasionally infests those from the Cucurbitaceae family, rendering it a significant vegetable pest.

Comment 5: Line 72: The hypothesis and a short idea of the methods should be stated clearly here. This could vastly improve the readability of the manuscript.

Reply: Thank you for your suggestion. In this revision, we have added a brief explanation regarding the hypothesis and methods. The details are as follows (Line80-84):

In previous studies, Physalis Plants have been shown to possess the ability to affect the immune systems of insects, leading to their death and impacting reproduction. To explore whether it has similar effects on H. vigintioctopunctata, we fed H. vigintioctopunctata larvae with leaves from four plants, including P. floridana Rydb., and conducted transcriptome sequencing to investigate any differences and changes.

Comment 6: Line 85: The lines of plants fed to insects is very confusing when presented in the journals format of having results before methods (instead of saying germ lines talk about the species of plants). 

Reply: Thank you for your advice. In this revision, we have added an explanation regarding the codes and groupings at the beginning of section 2.1. This should make the results less confusing to read and indeed improves the overall readability. The details are as follows (Line98-103):

All H. vigintioctopunctata larvae were divided into four groups and fed leaves from four different plants. The group fed Solanum nigrum L leaves is called the CK group; the group fed P. floridana leaves is called the P106 group; the group fed Solanum tuberosum L. leaves is called the POT group; and the group fed Solanum lycopersicum L. leaves is called the TOM group. Among these four groups, the CK group serves as the control group, while the other three groups are experimental groups.

Comment 7: Many minor things that need to be addressed and some odd phrasing. 

Reply: Thank you for your careful review and comments. We have corrected some language errors in this revised version。

Reviewer 2 Report

Comments and Suggestions for Authors

I am not really qualified in the transcriptomics field to be able to provide a comprehensive review. The paper is well written with interesting results that will allow the authors to home in on the immune system of the 28-spotted ladybird beetle.

Comments on the Quality of English Language

The English is fine. My main criticism is the references appear to be somewhat mixed up at the end of the manuscript at least. They need careful checking. In addition, the font on some of the figures is too small to be readable e.g. Figure 2, Figure 3, and Figure 5. There are also a number of minor errors listed below.

Paper by Xian-Zhong Wang et al: Comparative transcriptomics revealed Physalis floridana influences on the immune system off the 28-spotted ladybird beetle (Henosepilachna vigintioctopunctata)

General Comments

This paper is out of my comfort zone in terms of the science involved. It is generally well written, however. Unless there are serious flaws in the methodology, I recommend that it be accepted with minor revision.

The references need to be carefully checked as there are some at the end of the manuscript at least, that do not match with the references quoted.

Suggested corrections:

L15 significantly impacts

L70 omit ‘shortly’

L75 P. floridana

L104 ..images after eclosion at 3 days

Figures 2,3 & 5– Some of the font on these figures is too small to read

L151 D is an ‘upset plot’ – not sure what you mean.

L169 H. vigintiocpunctata

L216 ..using the host plant fed on as a feature.

L276 ..while only a few showed…

Fig 7 B Font size too small.

L313 .. in which 15 highly-expressed…

L316 David et al. proposed….[32] Reference 32 is not for David et al.

L332-333 Andrea et al is not the Reference 36 quoted or 37

L363 ..which were fed …  Omit ‘started being’

Author Response

Reviewer 2:

I am not really qualified in the transcriptomics field to be able to provide a comprehensive review. The paper is well written with interesting results that will allow the authors to home in on the immune system of the 28-spotted ladybird beetle.

Reply: Thank you for your meticulous review and invaluable comments on our manuscript. Your feedback has significantly enhanced the quality of our work. We have meticulously revised the manuscript according to your suggestions and have supplemented additional data where necessary. All modifications made in the revised manuscript are highlighted in blue for easy reference. Below, we offer detailed responses to the points you raised. In this revision, all changes are highlighted in blue font.

Comment 1: The English is fine. My main criticism is the references appear to be somewhat mixed up at the end of the manuscript at least. They need careful checking. In addition, the font on some of the figures is too small to be readable e.g. Figure 2, Figure 3, and Figure 5. There are also a number of minor errors listed below.

Reply: Thank you for your evaluation of our manuscript. Firstly, regarding the issue of references, there is no confusion in the citation; in fact, it was merely a matter of using authors' given names versus surnames. In this revision, we have integrated the reference names with those in the main text to ensure they look more correct. Secondly, concerning the font size in the figures, we have changed the font in this revision, applying a larger font to the latest figures. There may be parts where the font still cannot be adjusted to the largest size; this is a compromise for the layout and aesthetics of the figures.

Comment 2: This paper is out of my comfort zone in terms of the science involved. It is generally well written, however. Unless there are serious flaws in the methodology, I recommend that it be accepted with minor revision.

Reply: Thank you for your recognition for our work.

Comment 3: The references need to be carefully checked as there are some at the end of the manuscript at least, that do not match with the references quoted.

Reply: Thank you for your comments. In this revision, we have carefully reviewed all the references to ensure they are correctly cited in the appropriate places. Now, all references can be more easily matched with the main text content.

Comment 4: L15 significantly impacts

Reply: Thanks, it has been correct now.

Comment 5: L70 omit ‘shortly’

Reply: Thanks. It has been removed.

Comment 6: L75 P. floridana

Reply: Thanks! We have revised this error.

Comment 7: L104 ..images after eclosion at 3 days

Reply: Thank you for your advice. “at” has now been added.

Comment 8: Figures 2,3 & 5– Some of the font on these figures is too small to read.

Reply: Thank you for your suggestion. In this revision, we have tried to increase the font size of these three figures to make them clearer.

Comment 9: L151 D is an ‘upset plot’ – not sure what you mean.

Reply: UpSet is a graphical representation used for visualizing the intersections among multiple sets and can be considered an enhanced version of the Venn diagram. It is particularly suitable for displaying intersection scenarios when the number of sets exceeds five. This type of chart was introduced in the 2014 paper: 'UpSet: Visualization of Intersecting Sets' by the Visual Computing Group at Harvard Medical School, and it is a more readable data graph than Venn diagrams when dealing with multiple sets. It is a common picture type in transcriptome differential gene number analysis with multiple sample processing.

Comment 10: L169 H. vigintiocpunctata

Reply: Thank you. It's in italics now.

Comment 11: L216 ..using the host plant fed on as a feature.

Reply: Thanks! It has been revised now.

Comment 12: L276 ..while only a few showed…

Reply: Thanks. It is correct now.

Comment 13: Fig 7 B Font size too small.

Reply: Thanks. The font size has been enlarged.

Comment 14: L313 .. in which 15 highly-expressed…

Reply: Thanks. It is correct now.

Comment 15: L316 David et al. proposed….[32] Reference 32 is not for David et al.

Reply: Stanley, D. 's full name is Stanley Davied. This is our negligence. There is no problem with the citations in the article. We changed David to Stanley in L316 to make it easier for readers to match it with the references.

[32] Stanley, D.; Haas, E.; Kim, Y. Beyond Cellular Immunity: On the Biological Significance of Insect Hemocytes. Cells 2023, 12, 599.

Comment 16: L332-333 Andrea et al is not the Reference 36 quoted or 37

Reply: Andrea et al's article is reference 36. Like the previous article, there is no mis reference in this one, just the difference between the first name and the last name, which we have corrected to Barthel et al.

Comment 17: L363 ..which were fed …  Omit ‘started being’

Reply: Thanks. We have removed it.

Reviewer 3 Report

Comments and Suggestions for Authors

This study tested the effects of consuming Physalis floridana [a plant being investigated as a source of new, organic pesticides] on the transcriptome of Henosepilachna vigintioctopunctata [a notorious pest] as compared to feeding on several species of Solanum [including its natural host]. The insect expressed more detoxification genes and immunity genes when eating the former.

A big comment: Physalis floridana Rydb. is a synonym for Physalis pubescens L., accoridng to NCBI. https://ncbi.nlm.nih.gov/Taxonomy/Browser/wwwtax.cgi?mode=Info&id=300354  I'm not a botanist, but perhaps double check what the correct name is for this plant. You may need to change the species throughout the paper.

The science is sound. The methods are thorough. I am satisfied with the validity of the data.

The introduction is well written.

I personally prefer methods before results. The first paragraph of the results, for example, refers to codes like P106, CK, Pot, and Tom, but the reader won't know what these things are. Why have the methods after the results if the reader has to refer to the methods to understand the results? If the journal requirements allow it, either move the methods up to before the results, or indicate in the first paragraph of the results section what all these codes refer to. Code names, like abbreviations, must be defined in their earliest appearance in the manuscript, with no exceptions.
Similarly, in Figure 1, the caption should assign species to the code names. Captions should be understandable without referring to the main text.
Another option is, for the paper, to change the code names in every section and figure they appear to shorter versions of the species names: Pfl, Sni, Stu, Sly. The point of codenames is to make the paper easier to read, but in your case they make the paper significantly harder to understand.

In line 145, I would not say "all three groups" because you have four groups. What do those three groups have in common?

The first two paragraphs of the discussion are redundant with the introduction and can be deleted.

In the methods, the code names are not defined until section 4.4, even though P106 is used in section P106. This is not acceptable. Code names must be defined in their earliest appearance in the manuscript.

That immune pathways are triggered is a surprising find to me, though the introduction suggests it had been known, and the discussion states that in insects they have non-immune functions. I learned something new today,a nd that's a great sign for a paper! Still, can you confirm that the upregulation of the immune pathways in the P.floridana-fed insects is due to the plant itself, and not by a pathogenic bacteria that had been on those plants? At least add a line stating why, based on your methods, you believe this to be unlikely.

~~

Very minor typos, nothing requiring pre-acceptance editing. In a few points in the manuscript you forgot to italicize the species names (lines 75, 169). The abstract has a few errors:
-You have L. after the Solanum, but forgot the authority for P.floridana Rydb.  [though see my above comment on it being a synonym]
-Replace "significant impacts on the development and reproductive suppression of" with "significantly suppresses development and reproduction in"
-Delete "in contrast"

Author Response

Reviewer 3:

This study tested the effects of consuming Physalis floridana [a plant being investigated as a source of new, organic pesticides] on the transcriptome of Henosepilachna vigintioctopunctata [a notorious pest] as compared to feeding on several species of Solanum [including its natural host]. The insect expressed more detoxification genes and immunity genes when eating the former.

Reply: We are deeply grateful for the meticulous review and invaluable feedback you provided on our manuscript, which has significantly improved its quality. In response to your suggestions, we have comprehensively revised the document and incorporated additional data where appropriate. All revisions made in the updated manuscript are indicated in blue for clear identification. The following sections detail our responses to your comments and the corresponding adjustments we have implemented. Your comments are kept in black font, while our responses are highlighted in blue font for easy differentiation.

Comment 1: A big comment: Physalis floridana Rydb. is a synonym for Physalis pubescens L., accoridng to NCBI. https://ncbi.nlm.nih.gov/Taxonomy/Browser/wwwtax.cgi?mode=Info&id=300354  I'm not a botanist, but perhaps double check what the correct name is for this plant. You may need to change the species throughout the paper.

Reply: Thank you very much for your comment. The search result you found is correct; these terms are synonyms. They refer to two different names for the same plant. Both scientific names are correct and the relevant references are given below.

[1] Lu, J., Luo, M., Wang, L. et al. The Physalis floridana genome provides insights into the biochemical and morphological evolution of Physalis fruits. Hortic Res 8, 244 (2021). https://doi.org/10.1038/s41438-021-00705-w

[2] He C, Saedler H. Heterotopic expression of MPF2 is the key to the evolution of the Chinese lantern of Physalis, a morphological novelty in Solanaceae. Proc Natl Acad Sci U S A. 2005 Apr 19;102(16):5779-84. doi: 10.1073/pnas.0501877102. Epub 2005 Apr 11. PMID: 15824316; PMCID: PMC556287.

Comment 2: I personally prefer methods before results. The first paragraph of the results, for example, refers to codes like P106, CK, Pot, and Tom, but the reader won't know what these things are. Why have the methods after the results if the reader has to refer to the methods to understand the results? If the journal requirements allow it, either move the methods up to before the results, or indicate in the first paragraph of the results section what all these codes refer to. Code names, like abbreviations, must be defined in their earliest appearance in the manuscript, with no exceptions.

Similarly, in Figure 1, the caption should assign species to the code names. Captions should be understandable without referring to the main text.

Another option is, for the paper, to change the code names in every section and figure they appear to shorter versions of the species names: Pfl, Sni, Stu, Sly. The point of codenames is to make the paper easier to read, but in your case they make the paper significantly harder to understand.

Reply: Thank you for your suggestion. In this revision, we have added information about the codes and groupings at the beginning of section 2.1. This has improved the readability of the entire article. Your input is greatly appreciated! The added sentence is as follows (Line98-103):

All H. vigintioctopunctata larvae were divided into four groups and fed leaves from four different plants. The group fed Solanum nigrum L leaves is called the CK group; the group fed P. floridana leaves is called the P106 group; the group fed Solanum tuberosum L. leaves is called the POT group; and the group fed Solanum lycopersicum L. leaves is called the TOM group. Among these four groups, the CK group serves as the control group, while the other three groups are experimental groups.

Comment 3: In line 145, I would not say "all three groups" because you have four groups. What do those three groups have in common?

Reply: all three groups” have been corrected to “all three comparison sets”. Four groups were the treatment groups, in which the CK group was used as a control, while the other three groups served as experimental groups, and each experimental group was compared with the CK group as a comparison set. We have changed it for a clearer understanding.

Comment 4: The first two paragraphs of the discussion are redundant with the introduction and can be deleted.

Reply: Thank you for your comment. In this revision, the two paragraphs have been removed.

Comment 5: In the methods, the code names are not defined until section 4.4, even though P106 is used in section P106. This is not acceptable. Code names must be defined in their earliest appearance in the manuscript.

Reply: Thank you for your comment. This is indeed a serious error. In this revision, the initial explanation of these codes now appears at the very beginning of section 2.1 (at the start of the results section).

Comment 6: That immune pathways are triggered is a surprising find to me, though the introduction suggests it had been known, and the discussion states that in insects they have non-immune functions. I learned something new today, and that's a great sign for a paper! Still, can you confirm that the upregulation of the immune pathways in the P. floridana-fed insects is due to the plant itself, and not by a pathogenic bacteria that had been on those plants? At least add a line stating why, based on your methods, you believe this to be unlikely.

Reply: Thank you for your suggestion. In this study, all experimental plants were cultivated simultaneously in a constant temperature climate chamber with identical conditions, ensuring that all plants remained healthy. Therefore, the pathogens to which they were exposed were the same, and there would not be different pathogens on different plants leading to variations in the results. In this revision, we have added relevant descriptions in the Methods and Discussion sections. The specifics are as follows:

In this experiment, all plants were grown in the same environment to maximize con-sistency in their exposure to environmental factors, thereby reducing differences caused by pathogens on the plants. As a result, the activation of immune responses can be attributed to the plant hosts themselves rather than external pathogens. (Line333-336)

All test plants were grown in the same room using identical soil to ensure that they were exposed to consistent environmental factors. (Line362-364)

Comment 7: Very minor typos, nothing requiring pre-acceptance editing. In a few points in the manuscript you forgot to italicize the species names (lines 75, 169). The abstract has a few errors:

-You have L. after the Solanum, but forgot the authority for P.floridana Rydb.  [though see my above comment on it being a synonym]

-Replace "significant impacts on the development and reproductive suppression of" with "significantly suppresses development and reproduction in"

-Delete "in contrast"

Reply: Thank you for your careful review. All of the above issues have been addressed in this revision.